# Surface Modification of UiO-66 on Hollow Fibre Membrane for Membrane Distillation

**DOI:** 10.3390/membranes13030253

**Published:** 2023-02-21

**Authors:** Noor Fadilah Yusof, Amirul Afiat Raffi, Nur Zhatul Shima Yahaya, Khairul Hamimah Abas, Mohd Hafiz Dzarfan Othman, Juhana Jaafar, Mukhlis A. Rahman

**Affiliations:** 1Advanced Membrane Technology Research Centre (AMTEC), Universiti Teknologi Malaysia, Skudai 81310, Johor, Malaysia; 2Department of Control & Instrumentation Engineering, School of Electrical Engineering, Universiti Teknologi Malaysia, Skudai 81310, Johor, Malaysia

**Keywords:** metal–organic frameworks, UiO-66, hydrophobicity modification, membrane distillation

## Abstract

The hydrophobicity of metal–organic frameworks (MOFs) is critical in enhancing the separation process in membrane distillation. Herein, a new superhydrophobic University of Oslo 66 (UiO-66) MOFs was successfully constructed on the top of alumina hollow fibre (AHF) membrane for desalination purposes. The fabrication methodology of the membrane involved in situ growth of pure crystalline UiO-66 on top of AHF and post-synthetic modification by fluorosilane grafting. The resultant membrane was characterised to study the physicochemical properties of the pristine and modified membrane. A superhydrophobic UiO-66 with a contact angle of 163.6° and high liquid entry pressure was obtained by introducing a highly branched fluorocarbon chain while maintaining its crystallinity. As a result, the modified membrane achieved 14.95 L/m^2^∙h water flux and 99.9% NaCl rejection with low energy consumption in the direct contact membrane distillation process. Furthermore, the high surface energy contributed by UiO-66 is maximised to produce the maximum number of accessible sites for the grafting process. The synergistic effect of surface hydrophobicity and porous UiO-66 membrane in trapping water vapour shows great potential for desalination application.

## 1. Introduction

The global freshwater demand is likely to increase because of the increasing population growth, industrialisation, climate change, and contamination of accessible freshwater resources. Natural resources are not sufficient to meet that demand. One of the strategies to overcome water shortage is via seawater desalination. Desalination technology has been brought to attention as an alternative for freshwater production. It is increasingly becoming a competitive solution for providing freshwater. According to the International Desalination Association, the total number of desalination plants has reached 18,214 by 2020. Over 300 million people worldwide rely on desalinated water for some or all their daily needs [1]. Desalination has turned into a need to conquer water scarcity in numerous regions all over the world. Desalination is a process that converts salty water to freshwater by removing or reducing dissolved salt content. Desalination could provide sufficient potable water to sustain large populations living along the coast. For over 10 decades, numerous technologies have been developed for desalination. The technologies used in the industry generally can be classified into thermal process and membrane process. The most effective technology in the desalination process should undoubtedly be based on the membrane. Membrane separation has grown as a promising and well-known technology in water treatment. The materials used for membrane fabrication are of utmost importance since they influence heat and mass transfer across the membrane. The benefits of the membrane treatment process have been highlighted as having low energy consumption and being environmentally friendly. The advancement of membranes started in the 1960s when the first reverse osmosis-based water desalination plants were designed and installed [2]. The available processes involved in membrane desalination are reverse osmosis (RO) [3,4,5], forward osmosis (FO) [6,7], microfiltration (MF) [8], ultrafiltration (UF) [9], nanofiltration (NF) [10], and electrodialysis [11,12]. Different innovations such as membrane distillation (MD) have been explored because of their superiority regarding low operation energy, simplicity, and ability to be combined with solar energy [13,14], combining heat and power production or polygeneration [15]. MD is an advanced process that has been effectively adapted for water desalination and other industrial water treatment processes.

The selection of membrane materials is also strongly associated with the type of process used. Polymeric membranes are considered the first membrane generation, and they are on the frontline of wastewater treatment [16]. Inorganic materials such as ceramics have also been utilised due to their robustness of thermal and chemical stability. However, in recent years, the choice of membrane material has shifted to hybrid inorganic–organic materials. The incorporation of nanomaterial into the support has led to the development of novel membrane materials. The fabrication of versatile nanostructures has led to a wide range of applications, including water treatment. Different innovative synthesis and modification strategies have been developed to improve the capabilities of the desalination membrane. The incorporation of nanoparticles such as graphene [17], carbon nanotube [18], metal oxides [19], and many more has proved them to be excellent materials for membrane desalination. Today, porous crystalline materials, known as metal–organic frameworks (MOFs), have gained momentum in various research areas since their discovery in 1989 [20]. MOFs are a new class of coordination polymers constructed by the self-assembly of inorganic metal ions and organic ligands. The framework formation by strong coordination bonds demonstrates the chemical stability of the MOFs structure. MOFs possess outstanding characteristics with the unique property of high internal surface area (extending beyond 6000 m^2^/g) and ultrahigh porosity (up to 90% free volume) [21]. These fascinating features are the result of the ordered distribution of pore cavities with the majority of pore diameters being less than 1 nm. Therefore, these nanoscale cavities would fit the atoms and potentially be useful for gas storage, capture, and separation. The tunable pore sizes and well-defined molecular adsorption site properties [22] make MOFs a potential host for many guest molecules that can be entrapped within the void/cavity region formation. The degree of tunability can be constructed with various metal clusters and organic linkers, thus generating porous materials with a high surface area [22].

Countless groups over the world are synthesising and utilising MOFs in a variety of applications. Among the MOFs family, Zr-based MOFs have drawn great attention for its versatility. UiO-66 (University of Oslo, Zr_6_O_4_(OH)_4_(BDC)_6_) was synthesised by Cavka et al. in 2008 [23]. UiO-66 is a zirconium-based MOFs constructed by combining [Zr_6_O_4_(OH)_4_] with organic linker 1,4-benzodicarboxylic acid (BDC) to form a building block. The BDC linkers are coordinated to hexanuclear Zr_6_O_4_(OH)_4_ clusters. Each cluster is composed of twelve carboxylate groups coordinated to the Zr metal. The UiO-66 produces two microporous systems characterised by two void regions in the framework: octahedral and tetrahedral cage. These preferred cages have shown great attraction for the interaction of the guest molecules. The connection between clusters forms a face-centred cubic (fcc) lattice. The discovery of UiO-66 has resulted in intensive research in the uses of Zr-based MOFs for membrane-based water treatment. UiO-66 has also shown a great thermal stability when heated up to 450 °C or when exposed to high humidity conditions for several days, attributed to the strength of the carboxylate-Zr bond and the high connectivity of the metal clusters, Zr_6_O_4_(OH)_4_(CO_2_)_12_, which coordinated by twelve benzenedicarboxylate ligands in the ideal structure [24]. Despite the excellent performance of the UiO-66 active layer, the membrane wetting always becomes an unavoidable limitation to be used in the MD process since UiO-66 cannot act as hydrophobic property. Moreover, the selection of membrane materials is also strongly associated with the type of process to be used. In MD, the treatment process is based on the difference in vapour pressure as the driving force for separating the solutes. Compared with other treatment processes, MD possesses advantages in desalination application, yields high salt removal, and has the potential of dealing with highly concentrated saline water while utilising low feed temperature. However, the surface wettability of the membrane is a primary consideration during its selection and construction. Hydrophobic materials present a boundary for the liquid phase, transferring the vapour phase to pass through the membrane pores. Therefore, the surface modification of the membrane is one of the preferred methods to prevent membrane wetting by functionalising the membrane surface, for example, by silane grafting approach [25,26,27,28].

In this research, the synthesis of UiO-66 MOFs onto alumina hollow fibre (AHF) was designed to meet the criteria for liquid separation in the MD process. The excellent potential of UiO-66 has been adapted in liquid separation processes with different membrane preparation methods and separation processes [29,30,31,32,33]. However, minimal literature is found on the MD process due to the hydrophilic nature of UiO-66. Therefore, the modification of UiO-66 with fluoroalkylsilane agent (1*H*,1*H*,2*H*,2*H*-perfluorooctyltriethoxysilane, abbreviated as FAS) was constructed for the first time, to the best of our knowledge. This fabrication aimed to confer hydrophobicity to the UiO-66 membrane and potentially trigger a great performance in the MD process due to the integration of the respective advantages. Even though direct grafting of silane agent onto AHF is possible, the sintering temperature causes pore size and porosity reduction in AHF [34], thus reducing the active site, which reduces the effectiveness of the grafting process. The strategy to introduce UiO-66 before the grafting process aims to improve the surface energy, which will increase the hydrophobicity of the membrane surface. This work focused on the fabrication of new hybrid UiO-66/FAS membrane for desalination application. The physical characterisations and chemical interactions of UiO-66/FAS were explored, and the membrane separation performance in MD system was evaluated. It is assumed that the electrostatic and hydrophobic interactions possessed by the modified membrane enhance the transportation of water across the membrane.

## 2. Experimental

### 2.1. Materials

Commercially available alumina powder with three different particle sizes were purchased from Alfa Aesar: 1 µm (alpha, 99.9% metals basis, surface area 6–8 m^2^ g^−1^), 0.05 µm (gamma-alpha, 99.5% metals basis, surface area 32–40 m^2^ g^−1^), and 0.01 µm (gamma-alpha, 99.8% metals basis, surface area 100 m^2^ g^−1^). *N*-Methyl-2-pyrrolidone (NMP, solvent) and *N*,*N*-dimethylformamide (DMF, solvent) were purchased from QRëC, while polyethyleneglycol-30 dipolyhydroxystearate (Arlacel P135, dispersant) was purchased from Uniqema. Polyethersulfone (PESf, binder polymer) was supplied by Ameco Performance. Zirconium(IV) chloride (ZrCl_4_) was supplied from Acros. 1,4-Benzenedicarboxylic acid (H_2_BDC) and 1*H*,1*H*,2*H*,2*H*-perfluorooctyltriethoxysilane (FAS) were purchased from Sigma-Aldrich (St. Louis, MO, USA). Citric acid (99%, Fisher Chemical, Waltham, MA, USA), acetic acid (99%, HmbG, Hamburg, Germany), ethylene glycol and ethanol (99.9%, Merck, Rahway, NJ, USA) were obtained.

### 2.2. Fabrication of Alumina Hollow Fibre Membranes

The phase inversion-based extrusion and sintering process was used to prepare AHF membrane [35,36]. First, the concentrated ceramic dope was prepared by mixing alumina oxide powder in three different particle sizes with NMP and Arlacel in a mass ratio of 7:2:1. The suspension was then rolled using a planetary ball mill (NQM-2) for 48 h. Then, PESf was added to the suspension and the milling process was extended for another 48 h. After completing the milling process, the suspension was degassed to remove air bubbles. Next, the spinning method was conducted by extruding the ceramic suspension into a tube-in-orifice spinneret with an orifice diameter and inner diameter of 3 mm and 2.8 mm, respectively. The extrusion flowrate was fixed for 9 mL/min while the bore fluid flow rate was 10 mL/min. The resulting hollow fibre precursors were drawn into a coagulation bath with an air gap of 15 mm between the spinneret and coagulant bath. After the fabrication process, the extruded precursor fibre was immersed in tap water overnight to ensure the completion of the phase inversion and dried at room temperature. Upon the completion of immersion, the precursors were removed from the tap water, straightened, dried, and cut into 40 mm-long pieces. Finally, the resulting AHF membranes were sintered using a tubular furnace at 1400 °C.

### 2.3. Functionalisation of Alumina Hollow Fibre Membranes

#### 2.3.1. Seeding Growth of ZrO_2_

The outer surface of AHF was modified by seeding the growth of ZrO_2_ as the initial formation of UiO-66 nuclei. The seeding treatment was conducted by the sol–gel Pechini technique [37]. First, ZrCl_4_ was dissolved in 100 mL of distilled water with continuous stirring at 70 °C. Next, citric acid and ethylene glycol were added to the solution dropwise in a molar ratio of 1.2:1. The solution was aged at room temperature for 10 h. The AHF membrane was sealed off at both ends using polytetrafluoroethylene tape to avoid oxide formation inside the AHF. The membranes were then immersed horizontally in the ceramic boat containing Zr sol. Then, the membrane condition was calcined in a furnace at 400 °C for 1 h to grow ZrO_2_ on the membranes. Finally, the membranes were dried in an oven at 100 °C for 1 h.

#### 2.3.2. In Situ Solvothermal Synthesis of UiO-66

The continuous 0.1 M UiO-66 crystalline layer was prepared using the coordination modulator method by the in situ solvothermal technique [38]. First, the seeded AHF membranes were immersed in the reaction solution containing 180 mL DMF solvent, ZrCl_4_ and H_2_BDC linker. The metal precursor-to-organic linker mol ratio was 1:1. The mixture was then added with 20 mL of acetic acid. Then, the solution was poured into a gastight Teflon bottle with the AHF membranes fully immersed horizontally inside the bottle. The solvothermal process took place in an oven at a temperature of 120 °C for 24 h. When the reaction was complete, the content of the autoclave was cooled to room temperature. Finally, the membranes were washed with methanol to remove impurities and excess DMF solvent, and dried in an oven at 60 °C for 2 h. Meanwhile, the white precipitation was obtained by centrifugation at 8000 rpm for 15 min, washed with methanol, and dried.

#### 2.3.3. Hydrophobisation of Membrane

The hydrophobisation of UiO-66 membrane was conducted using silane surface modification by the immersion grafting technique [39]. The membranes were immersed in a solution containing ethanol and water in a ratio of 1:2 for 24 h. Upon completing the first immersion, the membranes were immersed in FAS solution in ethanol for 24 h to enable the grafting process of FAS with the membrane surface. Then, the membranes were rinsed and dried at 100 °C in an oven for 12 h. In this work, silane-modified UiO-66 membranes were prepared with different FAS concentrations of 2, 4, 6, 8, and 10% *v/v*, and the resulting membranes were designed as S1, S2, S3, S4, and S5, respectively (S = UiO-66-treated). Meanwhile, the controlled samples for S1, S2, S3, S4, and S5 were prepared in the same manner using a direct coating to the support alumina fibre (without UiO-66 layer) and they were denoted as U1, U2, U3, U4, and U5, respectively (U = UiO-66-untreated). The schematic illustration of prepared membranes can be seen in Figure 1.

### 2.4. Characterisations

The synthesised membranes were characterised to observe their physical and chemical properties. Morphological structures and surface roughness of membranes were observed using field emission scanning electron microscopy (FESEM, Zeiss Crossbeam 340). The samples were placed on metal stubs and sputter-coated with an ultrathin layer of Au under vacuum. Energy-dispersive X-ray spectroscopy (EDX) analysis coupled with FESEM was performed to determine the presence of Zr element, and the mapping technique was used to observe the distribution of Zr and other elements at the micrograph spots. In addition, atomic force microscopy (AFM, SPI3800) was used for the topography information, such as in determining surface roughness (roughness average, R_a_). The polycrystallinity properties of UiO-66 particles were identified by powder X-ray diffraction (XRD) (SmartLab X-ray diffractometer). The porosity properties of samples were measured by N_2_ adsorption–desorption isotherms performed by surface area and porosity analyser (ASAP 2020). The Brunauer–Emmett–Teller (BET) and Barrett–Joyner–Halenda (BJH) methods were used to analyse the surface area and pore size distribution, respectively. The thermal stability was determined by thermogravimetric analysis (TGA, TAQ5000IR). The derivative thermogravimetry (DTG) was plotted simultaneously with TGA plot. The thermogram was recorded in a temperature range of 0–1000 °C.

Meanwhile, for surface wettability, a contact angle goniometer (OCA 15EC, Dataphysics, Charlotte, NC, USA) was used to quantify the contact angle of a water droplet on the membrane surface. The liquid entry pressure (LEP) measurements were conducted using a water permeation system. A membrane was placed in a stainless steel adapter with one side sealed off. Then, water was added at the outer side of the membrane while the pressure was gradually increased and monitored via pressure gauge (0.5 bar interval). The pressure value was recorded when the water droplet started to appear from the membrane, which can be considered as the LEP value, which is in accordance with the Laplace–Young as in Equation (1) [40]:(1)$$\mathrm{LEP}=\frac{-2B\gamma_{L}}{r_{\max}}\cos\theta ,$$
where *B* is a geometric factor of membrane pore, $\gamma_{L}$ is the surface tension of liquid, $r_{\max}$ represents the maximum pore radius of membrane, and *ϴ* is the water contact angle.

For the chemical characterisations, the structural fingerprint for the conformation between UiO-66 particles and FAS coated layer were characterised by infrared spectroscopy. The presence of functional groups was determined by Fourier transform infrared spectrometer (FTIR, PerkinElmer Frontier, Chiba, Japan) using KBr pellet technique. The spectra were recorded in the range of 4000 to 400 cm^−1^. Lastly, the detailed bond formation in the sample was characterised by X-ray photoelectron spectroscopy (XPS, Shimadzu, Kyoto, Japan, Ultra DHD). This equipment utilised monochromated Al/Kα radiation as an excitation source. XPS spectra were collected for different binding energies for C1s, O1s, Zr3d, F1s, and Si2p.

### 2.5. Filtration Performance in the Membrane Distillation System

Direct contact membrane distillation (DCMD) process was carried out to evaluate the desalination performance of fabricated membranes, using the installation schematically shown in Figure 2. The membrane was placed in a stainless steel union cross connector and the outer surface of the membrane was positioned in contact with the hot feed side and the temperature was fixed at 60 °C. Meanwhile, the temperature was maintained at room temperature for the permeate side. The permeation flow rate was set for 150 mL/min at 1 atm. The hot feed solution was pumped into the module’s shell side, and the permeate flow from the distillate was collected from the lumen side of the hollow fibre. The DCMD process required 8 h for desalinating 40,000 ppm (500 mL) of NaCl solution. The weight changes in the permeate solution were monitored using weighing balances connected to the data logger. The pure water permeation flux (J_W_) was determined using Equation (2):(2)$$J_{w}=\frac{V}{t\;\times \;A},$$
where J_W_ is the pure water permeation flux (L/m^2^∙h,), v is the volume of water collected (L) at the permeate side, which is the measurement based on the weight change, t is the treatment time (h), and A is the total surface area of AHF membrane (m^2^). Meanwhile, the salt rejection was evaluated by measuring the salt content using an ion conductivity meter (Eutech Cond 6+, Thermo Scientific, Waltham, MA, USA) in the permeate solution for the initial and final process. The percentage of salt rejection was calculated using Equation (3):(3)$$\mathrm{Rejection}\;(\%)=\left(1-\frac{C_{p}}{C_{F}}\right)\times 100\%,$$
where R (%) is the percentage rejection and C_P_ and C_F_ are the concentration of permeate salt and initial salt at the hot feed solution, respectively.

## 3. Results and Discussion

### 3.1. Properties of Alumina Hollow Fibre

Figure 3a shows the cross-section and outer surface of pristine AHF prepared using the phase inversion and sintering technique. The AHF used as a support for the UiO-66 membrane had outer and inner diameters of 1.47 and 0.86 mm, respectively, with a thickness of 0.30 mm. The FESEM micrographs show that an asymmetric structure of AHF consists of two outer and inner regions, as shown in Figure 3b. The sponge-like morphology has a significant role in the water permeation process. Finger-like voids appear within the regions in which the inner region shows smaller voids than the outer region. The formation of these finger-like voids with an average length of 65.8 and 48.3 µm for the outer and inner regions, respectively, was influenced by the solvent and non-solvent exchange rate from the ceramic suspension to the coagulation bath [41] and affected by viscous fingering phenomenon during phase inversion process [42]. Grains are present on the outer surface (Figure 3c) with various sizes and mostly spherical shapes, distributed in a non-uniform way. Uncovered spots can be seen at the surface, which could disrupt the filtration process. Hence, the second growth is required to enhance the effectiveness of the membrane process. For the porosity analysis, N_2_ physisorption isotherm plots in Figure 3e show that the membrane is of Type III, which indicates the existence of micropores. A BJH pore size distribution plot produced from derivative pore volume normalised to the pore diameter demonstrated the highest distribution at approximately 2–3 nm (Figure 3d). The quantitative analysis of BET provided a specific surface area of 2.73 m^2^/g with a surface roughness of 24.24 nm (Figure 3f). The AHF has a lower surface area that needs to be increased for the reaction sites and to facilitate heat and mass transfer within the pores.

### 3.2. Physical Characterisation of the Membranes

The microstructure of AHF membranes for all samples is shown in Figure 3. Based on these images for membranes S1–S5 (Figure 4), the average particle size of UiO-66 obtained on the top of AHF ranged from 50 to 60 nm. Compared to the standard size of UiO-66 particles (200–500 nm) synthesised by hydrothermal [29,43,44], a smaller size of particles was obtained in this study. This shows that the pre-seeding treatment of ZrO_2_ using the sol–gel Pechini method enables a reduction in UiO-66 particle size. A small number of metal nodes (Zr^4+^) were pre-attached to the membrane substrate before the fully metal nodes were introduced during the hydrothermal process. This unique strategy was studied by Lan et al. in preparing ZIF-8 [45]. They determined that the separating process of nucleation and growth could control the crystal size formation. The crystal size of ZIF-8 was reduced from 440 nm to 45 nm by changing the amount of metal node during pre-seeding treatment. Moreover, turning the MOFs particle to smaller size produces a high surface area/volume ratio, which increases the outer surface energy of the particles [46]. Meanwhile, a layer of UiO-66 can be observed from the cross-section image with a thickness ranging from 0.90 to 1.24 µm. However, after FAS were grafted on the surface, there was no substantial layer could be observed by FESEM images. Previous studies have demonstrated that after a grafting process, FAS layer was barely visible when FAS with 0.01 mol/L in hexane [47] and 2 wt% in ethanol was used [48]. For the UiO-66-untreated membranes (Figure 5), the AHF grains were clearly observed (image comparison to Figure 3c), in which a similar condition was obtained. The FAS was not revealed by a distinct layer on the AHF surface. Owing to the limited number of hydroxyl groups on the surface of sintered AHF, it is evident that the FAS molecules were only bound to the surface and did not polymerise to form an obvious layer that could be observed by morphological analysis [47]. However, the spectra generated by EDX analysis consist of peaks showing the composition of Si and F elements (Appendix A).

The characterisation of UiO-66 and UiO-66 grafted with FAS (denoted as UiO-66/FAS) powder samples was conducted for further analysis. The surface roughness of the as-prepared sample was analysed from the AFM images shown in Figure 6a. AFM images displayed show the roughness of UiO-66 and UiO-66/FAS were 69.68 nm and 18.78 nm, respectively. A reduction in the surface roughness indicates that FAS has been successfully grafted on the membrane. Moreover, this decrement is in accordance with some studies that establish the decrease in membrane surface roughness after membrane hydrophobisation [49,50,51]. In addition, the UiO-66 surface induced a high Cassie–Baxter stability, in which a rougher surface could help to obtain a non-wetting state where liquid does not penetrate the grooves on a rough surface and leaves air gaps [52]. This would facilitate the vapour in the trapping process to pass through the UiO-66 layer. Apart from that, the rougher surface presumably has a higher surface area, leading to an enhanced grafting process. This is supported by the higher BET surface area and increase in XRD peak intensities for UiO-66 (Figure 6b). This proves that the immersion grafting technique successfully coated a layer of FAS. The powder XRD analysis was conducted to analyse the crystallinity of the samples. In Figure 6b, the emergence of sharp peaks in the XRD patterns is evidence of a good degree of crystallinity in the prepared samples. The essential peaks at approximately 2θ = 7.28°, 8.42°, 11.90°, and 25.60° agree with the existing literature in relation to the topologies of UiO-66 [53,54]. The XRD pattern for UiO-66/FAS has almost the same reflections as that of UiO-66, confirming the high similarity of crystalline properties for both samples. It is worth mentioning that the modification with FAS did not destroy the crystalline structure of UiO-66 (Figure 6c). Moreover, no extra peaks were associated with FAS because of its amorphous nature.

Gas adsorption and porosimetry techniques were used to evaluate the porous material properties [55]. The N_2_ adsorption–desorption experiments were carried out to analyse the surface area and porosity, including pore size and pore volume. The physisorption isotherm in Figure 7 shows that both samples exhibited a similar trend of Type I isotherms, which is typical for microporous materials. Furthermore, the BET-specific surface area of UiO-66 was 1322.18 m^2^/g with 0.22 cm^3^/g pore volume, and the average pore size was mainly distributed at 1.93 nm. A smaller particle size analysed by FESEM images reflects the ultrahigh surface area obtained. Moreover, UiO-66 and UiO-66/FAS samples demonstrated a uniform pore size distribution compared to AHF sample shown in Figure 3d. The surface area and pore size decreased upon incorporating FAS onto the frameworks (Table 1). This reduction might be related to the slight pore-blocking onto UiO-66. The attributed data from the gas adsorption experiment affirmed that the pores formed within the membrane.

TGA was carried out to determine the thermal stability of the prepared samples. The TGA–DTG curves of UiO-66 and UiO-66/FAS are presented in Figure 8. The first weight loss appearing at 22–83 °C for UiO-66 and 25–108 °C for UiO-66/FAS was due to the desorption of water at the surface of the samples. For the second stage, a small weight loss of UiO-66 appeared at the temperature of 129 °C, representing unreacted acetic acid, meanwhile that which appeared at the 220 °C originated from residual DMF solvent. Moreover, the weight loss that appeared at the temperature 329 °C was assigned to the dehydration of the OH terminal from Zr_6_O_4_(OH)_4_ nodes to Zr_6_O_6_ [56]. Meanwhile, there was no obvious solvent weight loss in the UiO-66/FAS sample at the same temperature range caused by the FAS that was coated on the UiO-66 frameworks. The desorption of solvent molecules from frameworks is lesser due to the heating process that took place during FAS grafting. Then, the TGA curve of UiO-66 declines at approximately 410–610 °C with a weight loss of 22.67%, and at 455–610 °C with a weight loss of 20.14% for UiO-66/FAS, suggesting the decomposition of the organic linker in the frameworks [56]. The significant decline at 396–455 °C in UiO-66/FAS was attributed to the loss of fluoroalkyl chains of FAS molecules. After 610 °C, both materials showed a residue of 43.58% and 42.3% for UiO-66 and UiO-66/FAS, respectively, which were assigned to the remaining ZrO_2_.

### 3.3. Surface Wettability

Water contact angle analysis was performed to determine the wettability of the prepared membranes (Figure 9). Pristine AHF and UiO-66 displayed hydrophilic properties, in which the contact angle values of the water drops were ±20° and ±25°, respectively. Moreover, the LEP values obtained for pristine AHF and UiO-66 membrane (Table 2) were 1.0 and 1.2 bar, respectively, exhibiting relatively low LEP values. Membranes were easily wetted caused the minimum operating pressure (1 atm) to be close to their LEP values. The results indicate that surface modification is needed to prevent liquid mass transfer across the membrane. As expected, after hydrophobic treatment, a contact angle over 90° was achieved, which meets the requirements for a hydrophobic surface [57], thus it is worth to be used in MD system. The high-water contact angle was achieved when the UiO-66 membrane was coated with a high FAS concentration, which increased the membrane’s hydrophobicity. This property is beneficial to obtain a high-water flux in the MD system as it allows only water vapour to cross the membrane. Moreover, surface wettability is directly related to surface energy, and a more energetically stable surface results in a less wettable surface [58]. It is demonstrated that introducing UiO-66 (S1–S5) before hydrophobic grafting promoted an increase in surface energy compared to the untreated membranes (U1–U5). When the concentration of FAS increased, LEP values showed excellent wetting resistance (Table 2). A high LEP value is preferred to avoid pore wetting phenomenon, which adversely affects the MD process. To run MD process, the trans-membrane pressure should be lower than the LEP value. The results indicated that introducing a material with hydrophilic characteristics and high surface area (UiO-66) could offer high surface energy for FAS coating. Therefore, the membranes were capable of preventing water permeation from entering the lumen during MD process and able to withstand high pressure.

### 3.4. Chemical Composition and Interaction of Modified Membranes

Figure 10 compares the FTIR vibrational spectra for the FAS, UiO-66, and UiO-66/FAS. For the FAS spectrum shown in Figure 10a, the sharp peak at 2800 cm^−1^ depicts the C–H alkane stretch from the triethoxy group. The stretching vibration frequency of the C–F band was observed at 1200 cm^−1^ in the FAS spectrum, and it shifted to a slightly higher frequency in the UiO-66/FAS spectrum, where the peaks at 1239 and 1144 cm^−1^ represent symmetrical and asymmetrical C–F vibration, respectively [16,17]. Moreover, the peaks at approximately 1100 cm^−1^ in the FAS spectrum shifted to a higher frequency in the UiO-66/FAS spectrum and are attributed to Si–O–R vibrations. It is assumed that FAS has been attracted to the UiO-66 via Si position as shown in the proposed interaction. Moreover, asymmetric and symmetric O–Zr–O vibration at approximately 500–750 cm^−1^ stretching [59] in the UiO-66 spectrum also shifted to a higher frequency in UiO-66/FAS spectra, revealing that FAS had been grafted to the Zr–O on the Zr cluster. The broad absorption band in the region of 3400–3600 cm^−1^ corresponds to the stretching vibration of O–H. This broadening peak might also overlap with the sp^2^ C–H stretch from the aromatic ring and some free carboxyl groups. The characteristic peaks at 1700 and 1400 cm^−1^ present in the UiO-66 and UiO-66/FAS spectra represent the stretching of C=O and –COO–, respectively, where it is absent in the FAS spectrum.

Deconvolution of core region XPS spectra was conducted to identify the chemical composition of the UiO-66 before and after grafting. Figure 11 presents the high resolution XPS spectra for the pure UiO-66 crystal corresponding to C1s, O1s, and Zr3d. The additional spectrum exhibited peaks due to Si2p and F1s for the UiO-66/FAS. All the peaks appear to be in good agreement with the bonding species detected in both samples, which are in agreement with previously reported of UiO-66-related study [60]. Three peaks centred at 288.8, 285.1, and 284.5 eV (UiO-66 sample) and a slight shift to 289.4, 285.7, and 284.6 eV (sample UiO-66/FAS) can be observed in the C1s spectra corresponding to O–C–O, C–O, C–C, respectively. Additional binding energy appears at 291.5 eV for the spectrum UiO-66/FAS corresponding to C–F covalent bond [61,62]. The spectrum of O1s showed peaks at the binding energy of 533.2, 531.8, and 530.1 eV (sample UiO-66) and 532.9, 531.7, and 528.9 eV (sample UiO-66/FAS) corresponding to C–OH, Zr–OH and O–Zr, respectively. An additional peak at 526.5 eV for the UiO-66/FAS spectrum associated with Si–O bond. The fitting peaks of F1S spectra appear at 686.4 and 688.7 eV for C–F semi-ionic bond and C–F covalent bond, respectively [61]. Meanwhile, the deconvolution of Zr3d spectra resulted in four peaks associated to Zr–O–C, Zr–O, Zr 3d_5/2_, and Zr 3d_3/2_ located in the range of ~181.0 to ~185.0 eV [63,64]. The spectra showed that Zr element exists in the form of Zr^4+^ based on the Zr 3d_3/2_ and 3d_5/2_ peak analysis [65,66]. The binding energy values of Zr 3d_5/2_ and 3d_3/2_ were in the range from 180 to 183 eV [64] for both samples with energy separation values of 1.3 and 1.6 eV for UiO-66 and UiO-66/FAS, respectively. From the analysis, there was no reduction in Zr^4+^ species in the deconvolution of Zr3d spectra, suggesting that the bonding formation was not directly on Zr sites. Moreover, the different intensities for Zr–O in Zr3d spectrum can be observed in both samples. The intense Zr–O peaks in UiO-66/FAS samples suggested the increased amount of the species at the surface of modified UiO-66.

Referring to the proposed interaction in Figure 12, Zr–O species in the secondary building unit (SBU) of UiO-66 plays a role in attracting the bonding formation with the silane moieties. In detail, the reaction is initiated by hydrolysation of silane by three hydrolysable substituents (triethoxy group) to silanol (RSi(OH)_3_). Then, it is followed by a condensation reaction, in which FAS that contains the silanol group is now ready to combine with another group by elimination of water molecules. Next is the grafting process. The O–H sites (O–H in green colour) at the silanol group interact by forming a hydrogen bond with possible sites at UiO-66, which is the O–H group at the metal cluster (O–H in black colour). During heat treatment at the last process, a covalent linkage is formed between UiO-66 and FAS molecule with simultaneous water loss. For clarification, the O–H bond in blue represents the twelve linkers for each Zr-cluster connected to another by their carboxylate groups. The drawing only represents three linkers for each SBU.

### 3.5. Desalination Performance

The DCMD system was employed to evaluate the effect of FAS concentration to confer hydrophobicity to the UiO-66 membrane. The long-term performance test was run for 8 h, where a 40,000 ppm NaCl solution was introduced into the lumen of the AHF and distilled water as permeate solution was introduced on the outer surface of the membranes. First, the pristine AHF (denoted as pristine Al_2_O_3_) and UiO-66 membrane were tested through the DCMD performance test. As depicted in Figure 13, the rejection of pristine Al_2_O_3_ and UiO-66 was 31.44 and 35.45%, with water flux of 240.94 and 228.80 L/m^2^∙h, respectively. After deposition of UiO-66, the rejection slightly increased due to the increases in contact angle degree of the UiO-66 membrane sample (Figure 9). The hydrophilic properties of membranes cause a relatively poor performance in MD. The results obtained from these membranes are still unsatisfactory to be used in MD operation system due to the wettability of the permeable membrane.

Figure 14a shows the separation performance of the AHF membranes after modification. All the tested membrane performed very well in the DCMD system. By operating at optimum conditions (1 atm, feed tank temperature: 60 °C, permeate tank: room temperature), most of the membranes maintained 99% of NaCl content in the feed tank. These high rejection values show that FAS coating on the membrane surface only allowed water vapour to be transported across the microporous membranes. The membranes with UiO-66 (S1–S5) had higher NaCl rejection values (~99%), meanwhile the membranes that did not incorporate UiO-66 (U1–U5) demonstrated the salt rejection ranging from 95 to 98%. The salt rejections obtained were 60 to 70% higher than those of the pristine membrane and unmodified UiO-66 membrane. For the water fluxes obtained during the MD process, the membranes with UiO-66 and without UiO-66 show an increase in water fluxes with increasing concentration of FAS. Membranes of S1, S2, S3, S4 and S5 had NaCl rejection values of 99.0, 99.3, 99.4%, 99.9% and 98.6% had water fluxes of 2.65, 5.61, and 7.82, 14.95 and 10.96 L/m^2^∙h, respectively. A similar trend was observed in the untreated membranes: samples U1, U2, U3, U4, and U5 also demonstrated high salt rejection at 95.0, 95.9, 98.7, 98.0, and 98.3%, with water fluxes of 2.31, 4.84, 5.79, 8.67, and 9.96 L/m^2^∙h, respectively. Lower water fluxes obtained at the lower concentration of FAS were caused by the membranes that ineffectively withstand a concentrated feed solution. This is in accordance with the contact angle values of the samples, where it directly influences water vapour permeability. A higher contact angle results in higher salt rejection.

Furthermore, by comparing the performance of both batches, the untreated sample showed a slight reduction in salt rejection and water flux compared to the treated batch. From this observation, when the UiO-66 was introduced, the surface energy was maximised and produced the maximum number of accessible sites for the grafting process (Figure 15). Regarding the surface energy, three different sizes of sintered Al_2_O_3_ powder (0.01, 0.05, and 1 µm) were directly coated with FAS to observe the chemical interactions. Referring to the FTIR spectra (Figure 16), there is a weak absorbance in Si–O stretching, indicating fewer Si–O species in the sample, thus proving that it has a limited number of binding sites. Consequently, the presence of UiO-66 in the membrane has a significant effect on membrane performance, which can be correlated to the structural and physical characteristics of the membrane, such as specific surface area, porosity, and surface roughness. Apart from that, the incorporation of UiO-66 layer also led to substantial salt rejection because of its ability to form an electrostatic interaction with negatively charged NaCl.

In MD system, water vapour diffuses down by vapour potential gradient through the membrane and condenses at the inner side of the membrane surface, thus producing the permeate flux. The water permeate flux is commonly determined by the effect of pore size exclusion and interactions between the permeate and surface of the membrane. With respect to the unique structure of UiO-66, it has two sizes of the void region in the framework: octahedral and tetrahedral regions with diameters 11 and 8 Å, respectively, connected by a 6 Å triangular pore window [67]. It is comparable with the water size, 2.7 Å [68], which is smaller than 6 Å. Therefore, the water vapour can enter that window aperture and be trapped into the void regions. Meanwhile, the salt ions with diameter 7.2 Å (${\mathrm{Na}}^{+}$) and 6.6 Å (${\mathrm{Cl}}^{-}$) might be prevented or repulsed by the membrane. Water molecules can entrap on the two void regions of UiO-66 depending on the hydrogen bonding network and water loading [69]. This phenomenon has been studied comprehensively by various works that used UiO-66 as the effective material for gas application [70,71,72]. Therefore, UiO-66 has significant implications for use in the MD filtration system without compromising salt rejection.

Table 3 compares as-prepared UiO-66 membranes with other UiO-66/UiO-66-isoreticular membranes in desalination performance. This work agrees with the previous studies affirming that the UiO-66 membrane is particularly efficient for desalination test and enables almost 100% NaCl rejection. The results in this study demonstrate that UiO-66/FAS are compatible to be membrane’s material for the MD application.

## 4. Conclusions

The present study has successfully demonstrated the modification of AHF membrane by incorporating UiO-66 MOFs prior to the silane grafting process. A two-step synthesis strategy was undertaken to grow a continuous UiO-66 crystal on AHF using an in situ solvothermal technique followed by immersion grafting technique in FAS solution. The effect of FAS loading on the AHF substrate with and without growing the UiO-66 layer was investigated. The salt rejections were enhanced by 60 to 70% compared to that of the pristine and unmodified UiO-66 membrane. The best-performing UiO-66/FAS membrane showed a 99.99% NaCl rejection with a FAS loading of 8% *v/v* with a water flux of 14.95 L/m^2^∙h in DCMD performance test. Furthermore, the UiO-66 has high surface energy for FAS grafting, which is proved by its structural properties, such as surface area and roughness, porosity, and contact angle values. Therefore, this study aims to develop material with synergic effects contributed by MOFs and hydrophobic material integration in the MD process.

## Figures and Tables

**Figure 1 membranes-13-00253-f001:**
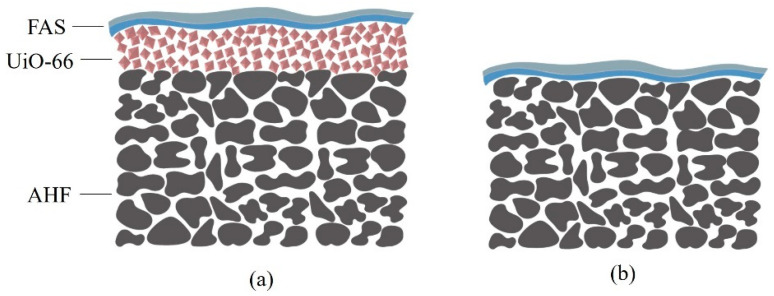
Schematic illustration of the prepared membranes (**a**) UiO-66-treated membrane (S1–S2) and (**b**) direct coated FAS onto the membrane (U1–U2).

**Figure 2 membranes-13-00253-f002:**
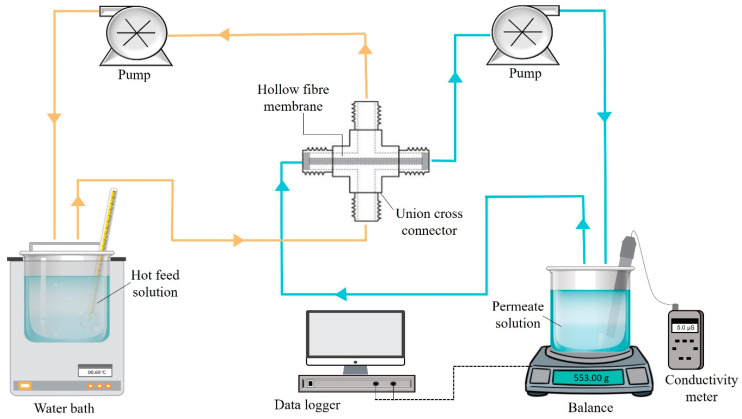
Schematic of DCMD experimental set-up.

**Figure 3 membranes-13-00253-f003:**
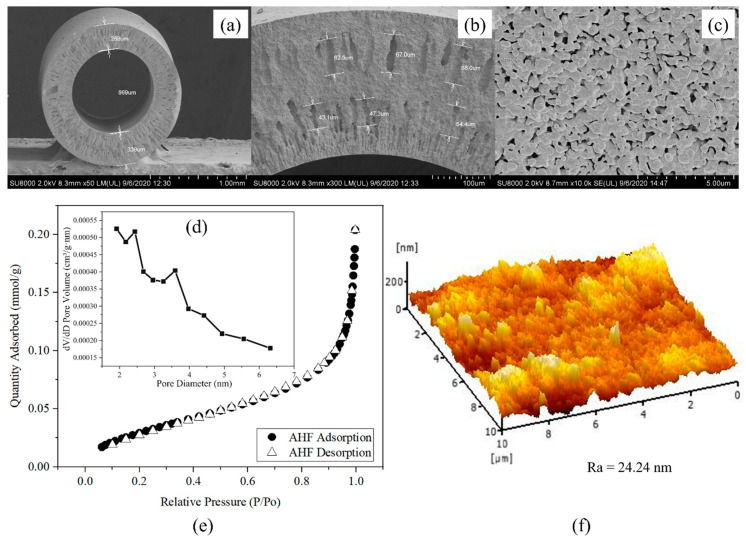
FESEM micrographs of AHF (**a**) cross-section, (**b**) enlarged cross-section, and (**c**) outer surface; (**d**) BJH porosity; (**e**) N_2_ adsorption–desorption isotherm; and (**f**) AFM image.

**Figure 4 membranes-13-00253-f004:**
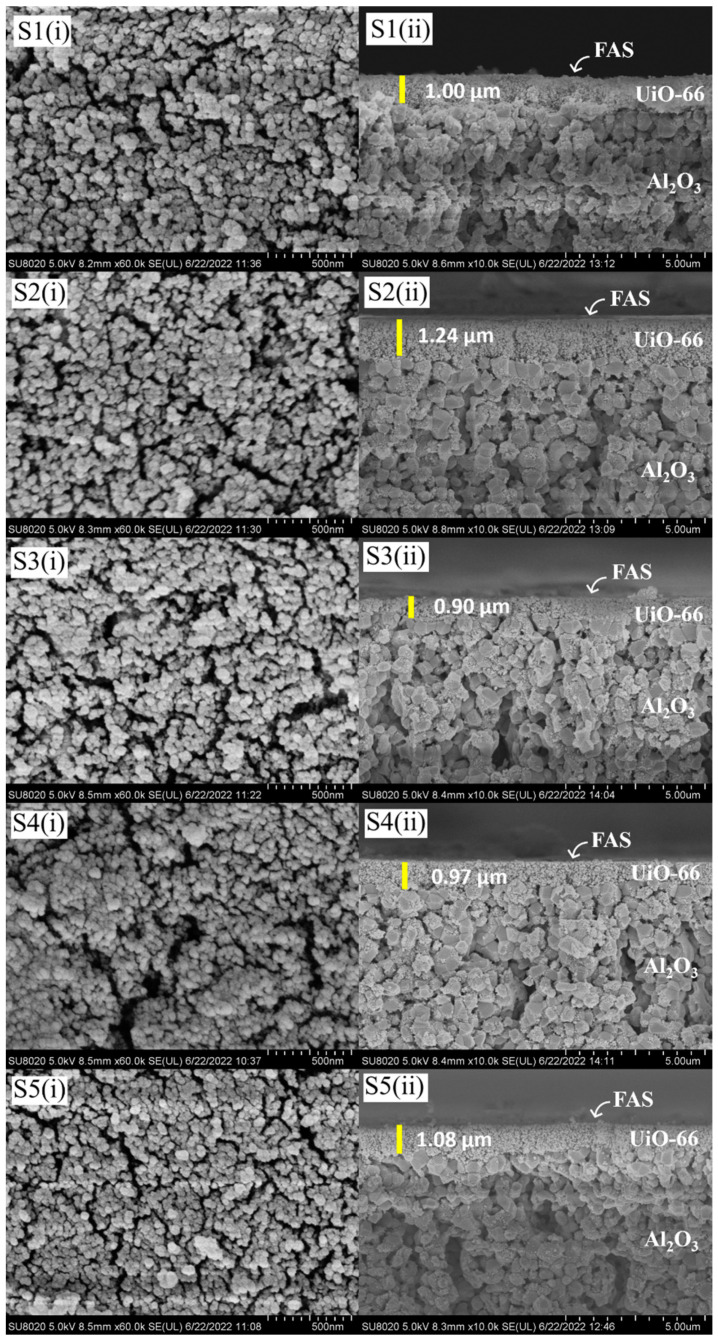
FESEM micrographs of S1–S5 membranes. (Numberings of (i) and (ii) are abbreviated for outer surface and cross-section image, respectively.)

**Figure 5 membranes-13-00253-f005:**
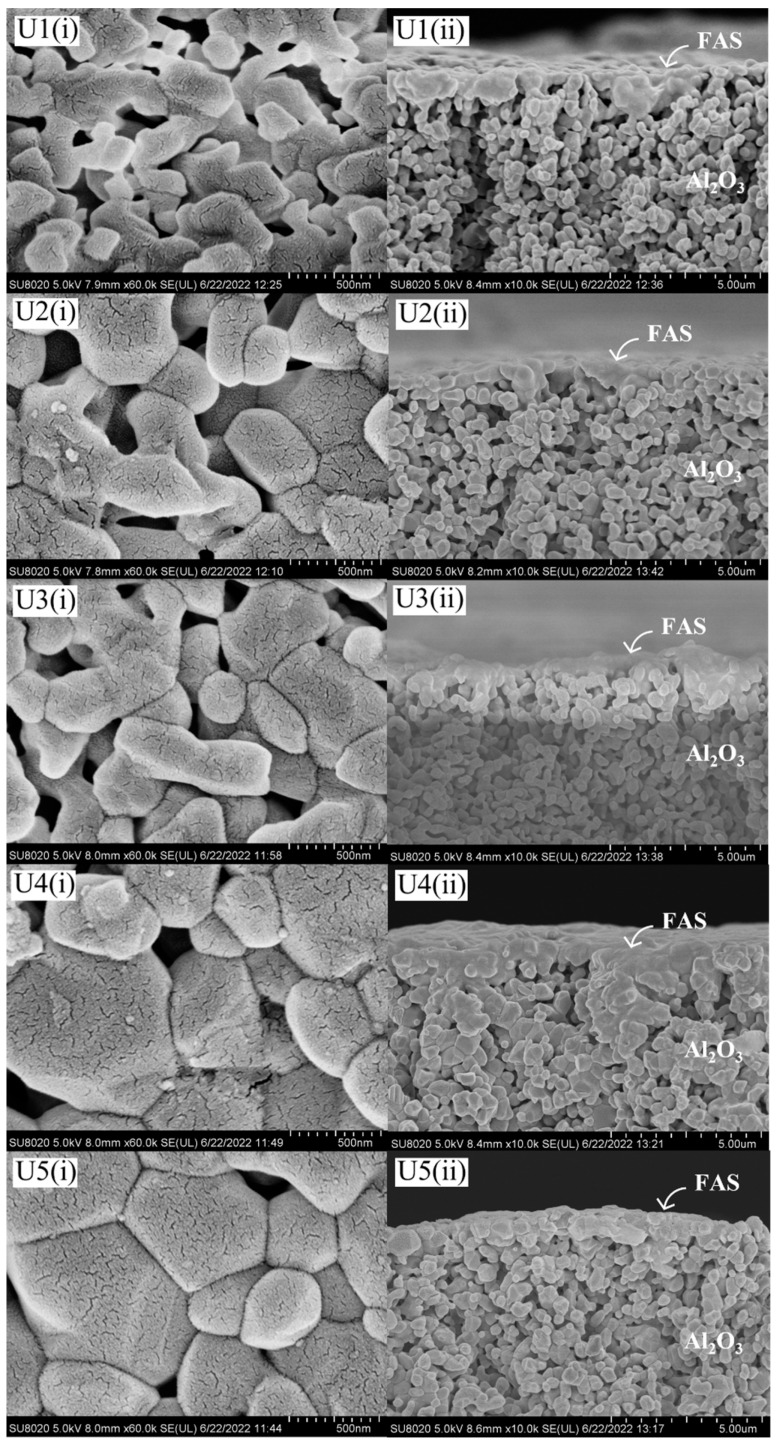
FESEM micrographs of U1–U5 membranes. (Numberings of (i) and (ii) are abbreviated for outer surface and cross-section image, respectively.)

**Figure 6 membranes-13-00253-f006:**
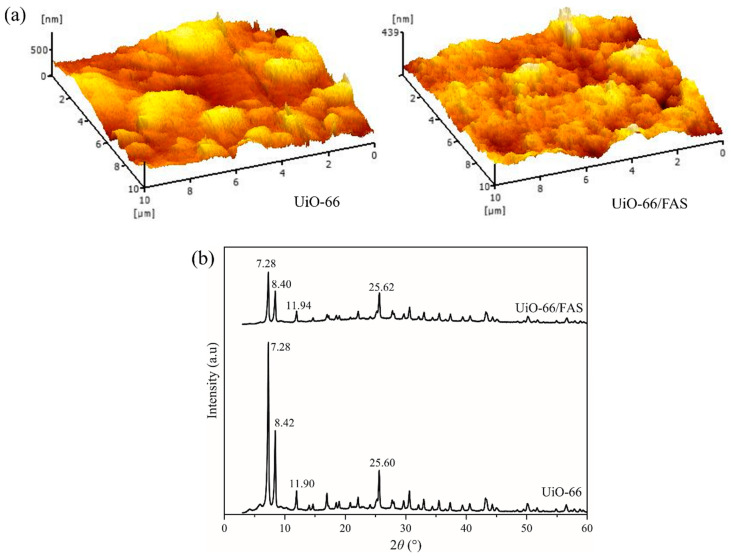
(**a**) AFM image, (**b**) XRD diffractogram of UiO-66 and UiO-66/FAS.

**Figure 7 membranes-13-00253-f007:**
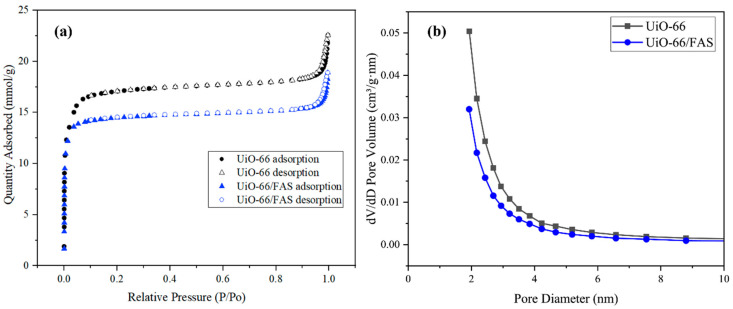
(**a**) N_2_ adsorption–desorption isotherm and (**b**) BJH pore diameter distribution.

**Figure 8 membranes-13-00253-f008:**
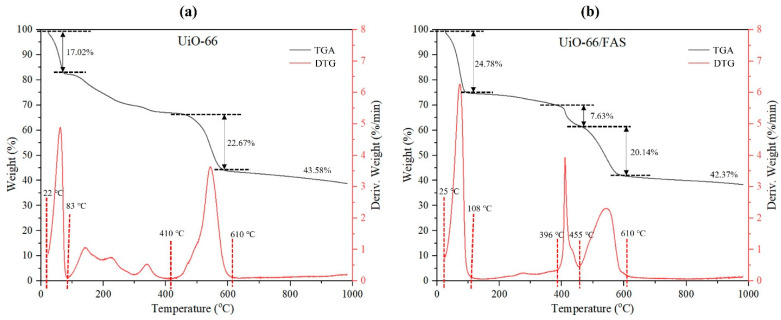
TGA-DTG thermogram of (**a**) UiO-66 and (**b**) UiO-66/FAS.

**Figure 9 membranes-13-00253-f009:**
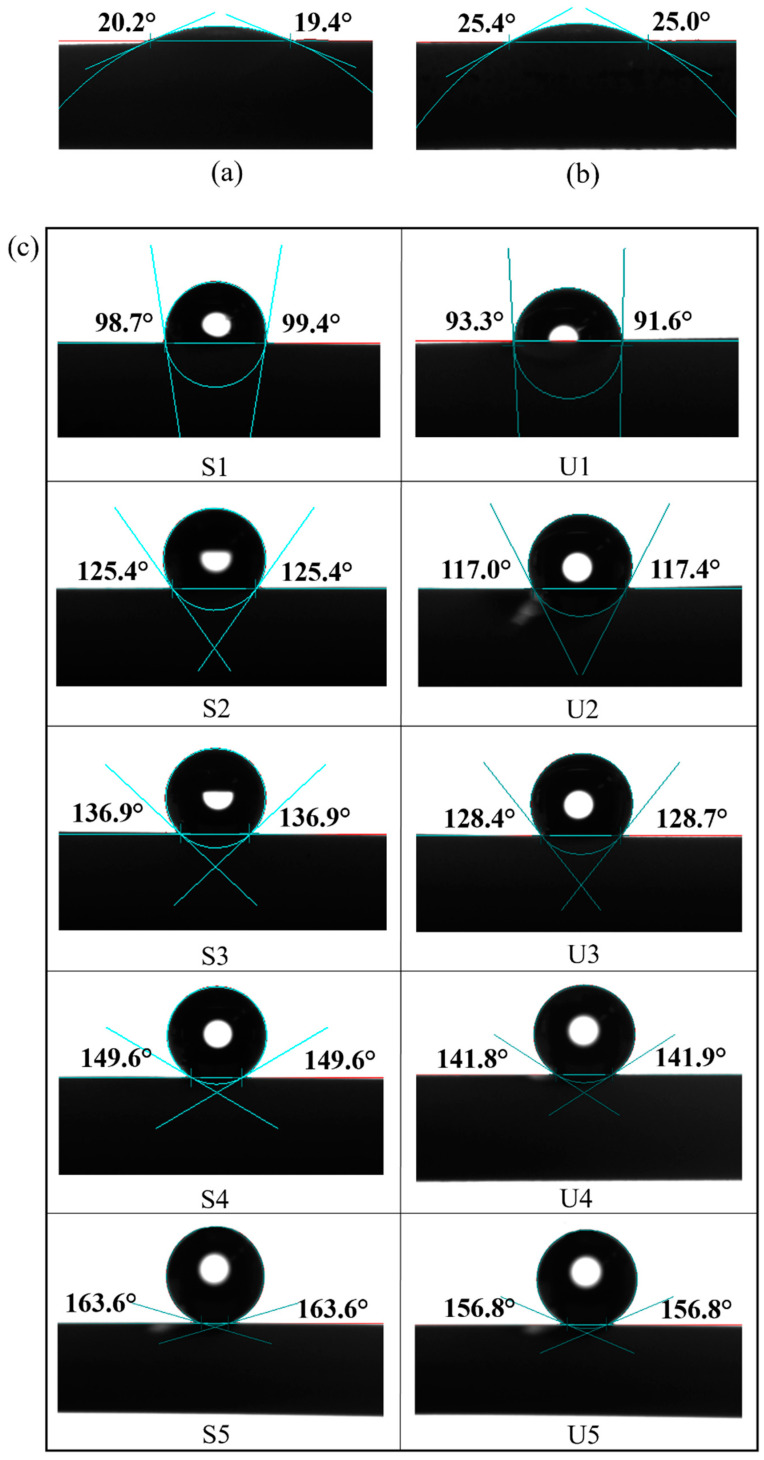
Water contact angle of prepared membranes of (**a**) pristine AHF, (**b**) UiO-66, and (**c**) modified membranes.

**Figure 10 membranes-13-00253-f010:**
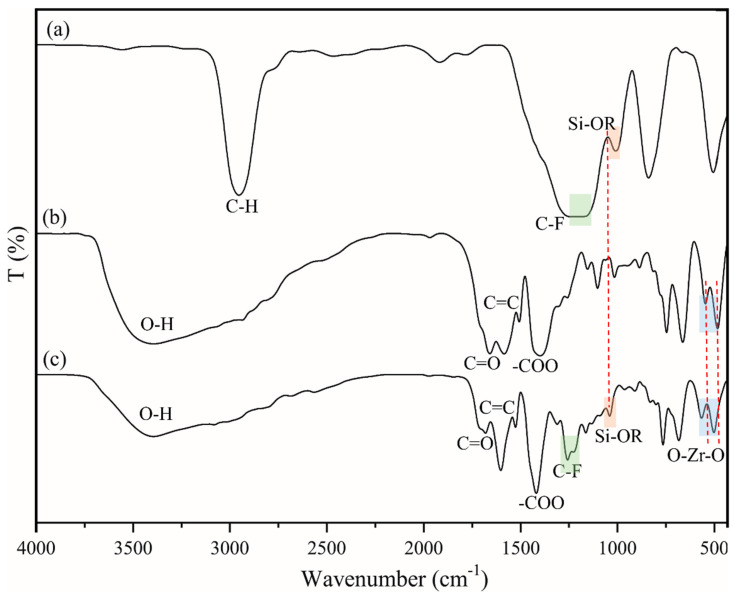
FTIR vibrational spectra for (**a**) FAS, (**b**) UiO-66, and (**c**) UiO-66/FAS.

**Figure 11 membranes-13-00253-f011:**
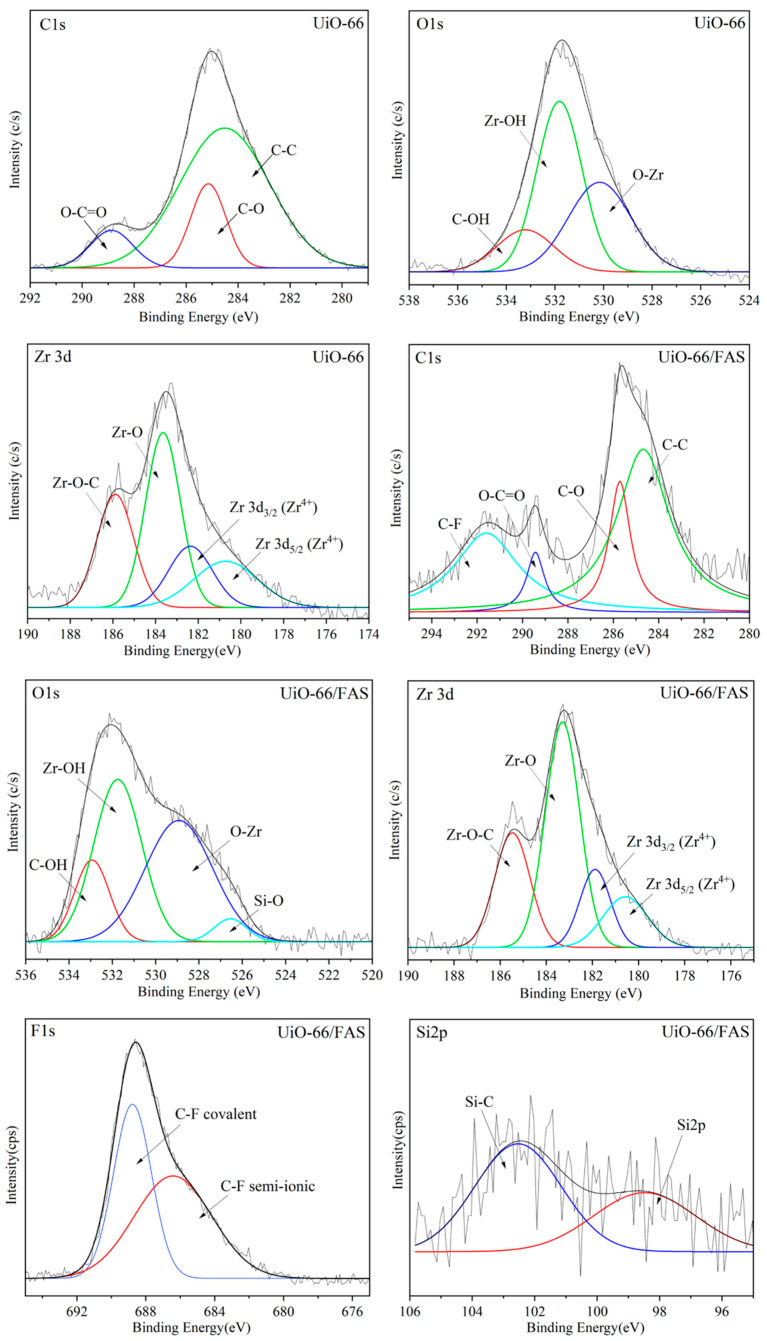
XPS spectra of UiO-66 and UiO-66/FAS.

**Figure 12 membranes-13-00253-f012:**
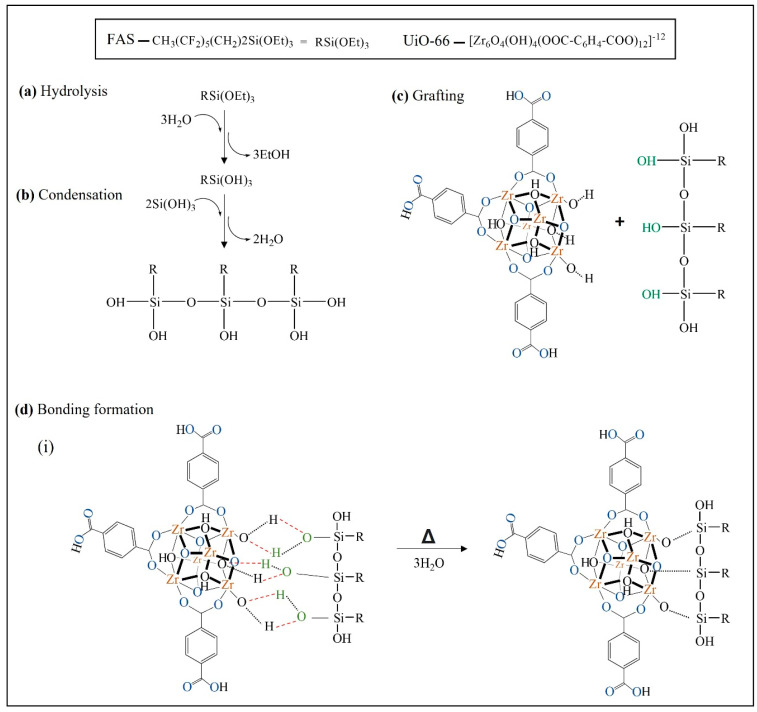
Proposed interaction mechanism of silane-grafted UiO-66 process.

**Figure 13 membranes-13-00253-f013:**
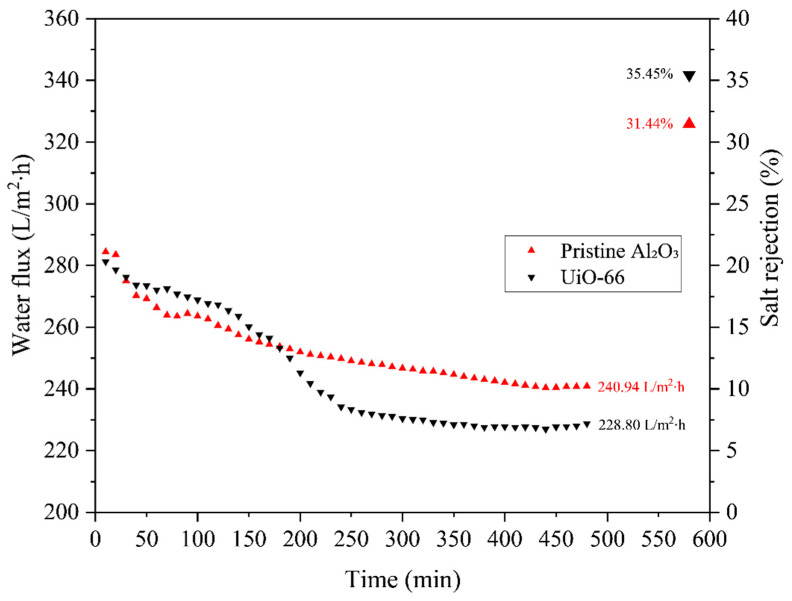
Water fluxes and salt rejections of lumen for the pristine Al_2_O_3_ and UiO-66 membrane before modification in 8 h DCMD operation.

**Figure 14 membranes-13-00253-f014:**
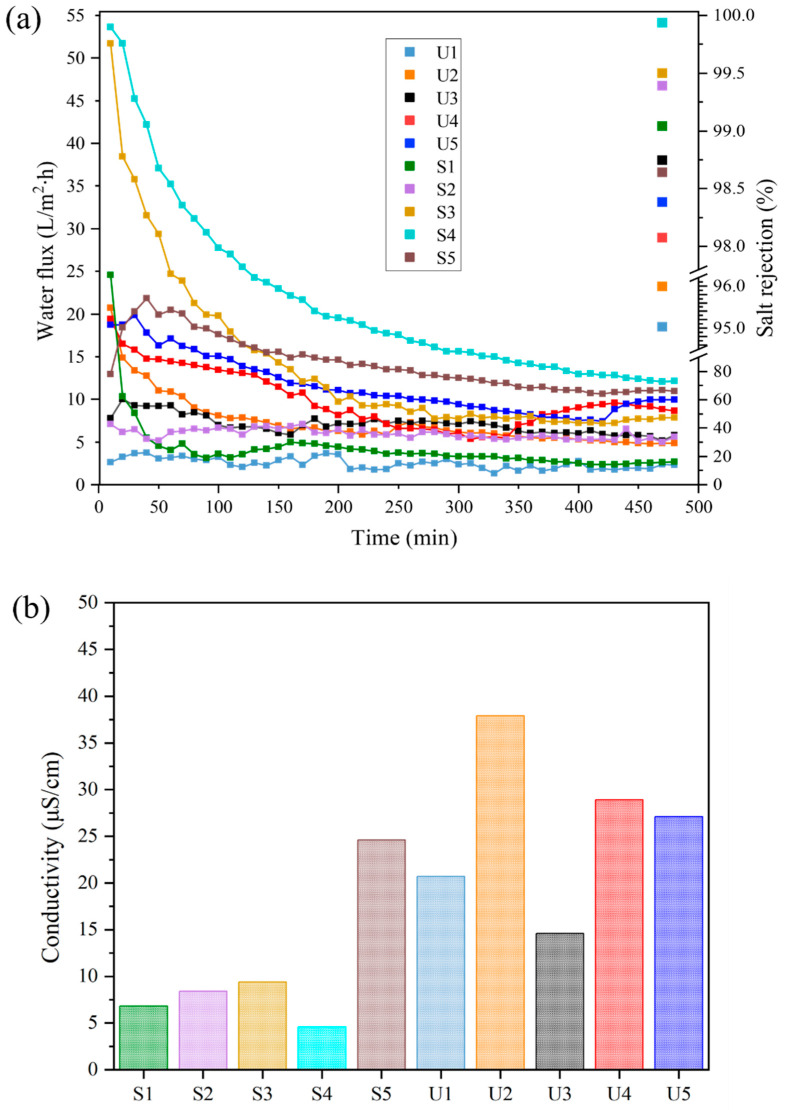
(**a**) Water fluxes and salt rejections, and (**b**) final conductivity values of lumen for the as-prepared membranes in a 8 h DCMD operation.

**Figure 15 membranes-13-00253-f015:**
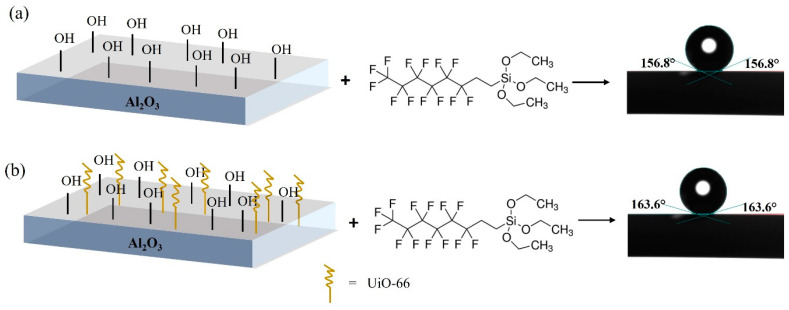
Schematic illustration of AHF modification by (**a**) AHF without UiO-66 and (**b**) AHF with UiO-66.

**Figure 16 membranes-13-00253-f016:**
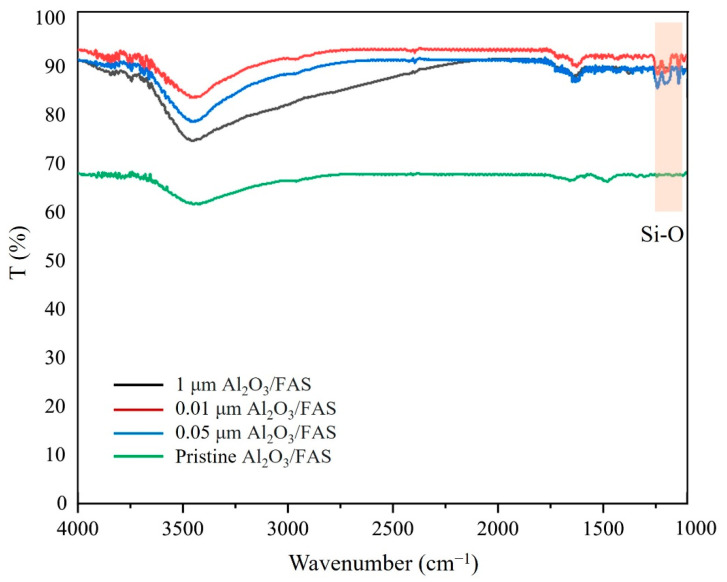
FTIR spectra for Al_2_O_3_ directly coated by FAS.

**Table 1 membranes-13-00253-t001:** Porosity analysis of UiO-66 and UiO-66/FAS.

Samples	BET Surface Area (m^2^/g)	BJH Pore Diameter (nm)	BJH Pore Volume (cm^3^/g)
UiO-66	1322.18	1.93	0.22
UiO-66/FAS	1094.09	1.92	0.17

**Table 2 membranes-13-00253-t002:** LEP values for synthesised membranes.

Membranes	LEP (bar)
Pristine AHF	1.0 ± 0.05
UiO-66	1.2 ± 0.1
S1	2.4 ± 0.05
S2	2.8 ± 0.1
S3	4.2 ± 0.1
S4	4.7 ± 0.05
S5	6.1 ± 0.2
U1	2.3 ± 0.05
U2	3.0 ± 0.05
U3	3.8 ± 0.3
U4	4.5 ± 0.1
U5	5.8 ± 0.1

**Table 3 membranes-13-00253-t003:** Comparison of UiO-66 membranes prepared in this study and the UiO-66 membranes in the literature reported in the desalination application.

MOFs	Membrane Process	Membrane Conditions	Water Flux	Salt Rejection (%)	Ref.
UiO-66	MD	Hybrid MOFs/FAS deposited on AHF	2.65 to 14.95 L/m^2^∙h	NaCl = 95–99	This work
UiO-66	Dead-end filtration	MOFs deposited on AHF	0.14 to 0.28 L/m^2^∙h∙bar	NaCl = 47	[29]
UiO-66-NH_2_	Reverse osmosis	MOFs incorporated on polyamide thin film nanocomposite	1.88 to 3.48 L/m^2^∙h∙bar, 1.74 to 2.99 L/m^2^∙h∙bar	NaCl = 95	[73]
UiO-66-NH_2_	Pervaporation	MOFs deposited on 3-aminopropy-ltriethoxysilane-modified Al_2_O_3_ tube	1.5 to 12.1 kg/m^2^∙h	${\mathrm{Na}}^{+}\;$= 99.94 ${\mathrm{Cl}}^{-}$= 99.98	[74]
UiO-66-NDC	Forward osmosis	MOFs coated with UV curable resin and deposited on AHF	6.0 to 16.18 L/m^2^∙h	NaCl = 80%	[75]

## Data Availability

Not applicable.

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
