# Peer review of "Surface Modification of UiO-66 on Hollow Fibre Membrane for Membrane Distillation"

_membranes, 2023, doi:10.3390/membranes13030253_

Round 1

Reviewer 1 Report

The introduction is too extensive and should be reduced considerably, highlighting mainly what is new to be done in the subject of membrane distillation.

The thermogravimetric analysis must include the theoretical and experimental percentages of the observed mass losses.

Linea 231 wavenumbers, must be removed.

Reviewer 2 Report

The paper is overall well designed and well written, with a good degree of detail. Nonetheless few point should be addressed:

- In introduction: MOFs are not polymeric materials, this term is misused. Polymeric refers to polymers and although the term coordination polymers exists, MOFs are crystalline materials and should not be referred as polymeric.

- There is no clear explanation why UiO-66 was the selected framework and not a MOF already displaying hydrophobic features. Few words justifying this choice should be added.

- page 4, line 180: solvent and not sovent.

- concerning the membrane 'synthesis': why was a DMF solvent method chosen when water treatment is the expected application? numerous water based synthesis for UiO-66 exist. It seems from TGA there is still DMF remaining in UiO-66 (the small loss after 200°C)

- Additionally, the TGA interpretation should be reviewed has the attribution of such a important loss from 80°C to 410°C cannot be assigned to water loss from the clusters, which normally do not contain coordinated water but rather terminal OH groups which do not leave easily the oxo-cluster. Moreover, DMF cannot leave before 83°C as its boiling point is 153°C, hence indicating that this loss can be assigned to DMF from particles surface it is not correct. The proposed reference do not claim that this loss is attributed to water loss from the clusters either but rather from remaining linker.

- Figure 1b) There is no UiO-66 on this membrane, please correct the figure caption as it may lead to an incorrect interpretation of the figure.

- What is the mechanical stability of the membranes? is there any brittleness or particle disaggregation once the membranes are shaped?

- concerning the membranes roughness (page 8) : the decrease in the membrane roughness is attributed to the hydrophobic properties, however this seems to be a bit over interpreted and a small rearrangement would be needed... In principle, when a coating is added to the surface of a membrane (hydrophobic or not), if the coating is homogeneous, the surface of the membrane will become smoother.

- After the Desalinization studies, is there any evidence that the MOF structure as remained stable? How about the membrane? any studies were made to ensure that there was no MOF leaching?
